# Targeted Therapy for Glomerulonephritis Using Arterial Delivery of Encapsulated Etanercept

**DOI:** 10.3390/ijms24032784

**Published:** 2023-02-01

**Authors:** Natalia A. Shushunova, Oksana A. Mayorova, Ekaterina S. Prikhozhdenko, Olga A. Goryacheva, Oleg A. Kulikov, Valentina O. Plastun, Olga I. Gusliakova, Albert R. Muslimov, Olga A. Inozemtseva, Nikolay A. Pyataev, Alexander A. Shirokov, Dmitry A. Gorin, Gleb B. Sukhorukov, Olga A. Sindeeva

**Affiliations:** 1Science Medical Center, Saratov State University, 83 Astrakhanskaya Str., 410012 Saratov, Russia; 2Institute of Medicine, National Research Ogarev Mordovia State University, 68 Bolshevistskaya Str., 430005 Saransk, Russia; 3Scientific Center for Translational Medicine, Sirius University of Science and Technology, 1 Olympic Ave, 354340 Sirius, Russia; 4Gene and Cell Therapy Laboratory, Pavlov University, 6-8 L’va Tolstogo Str., 197022 St. Petersburg, Russia; 5Institute of Biochemistry and Physiology of Plants and Microorganisms, Saratov Scientific Centre of the Russian Academy of Sciences (IBPPM RAS), Prospekt Entuziastov 13, 410049 Saratov, Russia; 6Center for Photonic Science and Engineering, Skolkovo Institute of Science and Technology, Bolshoy Boulevard 30, 121205 Moscow, Russia; 7A.V. Zelmann Center for Neurobiology and Brain Rehabilitation, Skolkovo Institute of Science and Technology, Bolshoy Boulevard 30, 121205 Moscow, Russia; 8Siberian State Medical University, 2 Moskovskiy Trakt, 634050 Tomsk, Russia; 9School of Engineering and Materials Science, Queen Mary University of London, Mile End Road, London E1 4NS, UK

**Keywords:** targeted drug delivery, etanercept, Enbrel, polyelectrolyte microcapsules, intra-arterial injection, Freund’s complete adjuvant, acute glomerulonephritis, nephritis

## Abstract

Complex immunosuppressive therapy is prescribed in medical practice to patients with glomerulonephritis to help them overcome symptoms and prevent chronic renal failure. Such an approach requires long-term systemic administration of strong medications, which causes severe side effects. This work shows the efficiency of polymer capsule accumulation (2.8 ± 0.4 µm) containing labeled etanercept (100 μg per dose) in the kidneys of mice. The comparison of injection into the renal artery and tail vein shows the significant superiority of the intra-arterial administration strategy. The etanercept retention rate of 18% and 8% ID in kidneys was found 1 min and 1 h after injection, respectively. The capsules were predominantly localized in the glomeruli after injection in mice using a model of acute glomerulonephritis. Histological analysis confirmed a significant therapeutic effect only in animals with intra-arterial administration of microcapsules with etanercept. The proposed strategy combines endovascular surgery and the use of polymer microcapsules containing a high molecular weight drug that can be successfully applied to treat a wide range of kidney diseases associated with glomerular pathology.

## 1. Introduction

The term glomerulonephritis includes a number of kidney diseases involving an abnormal immune process, tissue damage, and inflammation predominantly in the glomeruli. Ineffective therapy strategies lead to the rapid progression of this type of disease and to renal failure in the terminal stage. Although the onset of the initial pathological process is associated with different cells (mesangial cells, epithelial cells, and endothelial cells) for various types of glomerulonephritis (IgA nephropathy, focal and segmental glomerulosclerosis, infection-associated glomerulonephritis, and others) [1], the treatment of most diseases of such etiology is based on the use of immunosuppressive agents (mainly glucocorticoids) and has been for 70 years [2]. Very high doses of glucocorticoids are used at the first stage of treatment and reduced only considering positive therapy dynamics. A full course of therapy lasts about 6 months on average. The severity of the side effects is worrisome. In addition to painful external changes (weight gain, skin problems), severe physiological and metabolic complications occur (muscle atrophy, diabetes, hypertension, stomach ulcers, and bone loss) [3]. Immunosuppressive therapy carries an increased risk of infectious complications [4].

Glucocorticoids administered intravenously or orally circulate throughout the body and accumulate widely and non-specifically, meaning they gather not only in the area requiring therapy but also in healthy tissues, which leads to side effects. The idea of directing therapeutic agents to the site of pathology by implementing various types of carriers has already shown its effectiveness for the treatment of a number of experimental model diseases in vivo [5,6,7]. In addition to the use of appropriate carriers, the method of administration has a strong influence on the subsequent biodistribution of the therapeutic agent. Narute Tomita et al. showed that injection into the renal artery of liposomes with immobilized NF-kB decoy oligodeoxynucleotide has therapeutic potential in rapidly progressive glomerulonephritis [8].

Polyelectrolyte multilayer microcapsules are one of the promising systems for encapsulation and prolonged release of drugs at important sites [9,10,11,12,13]. There are a number of unique properties suitable for arterial targeting: high stability (important for cargo retention [14]), complete biodegradability (ensures the "purification" of vessels from the shells’ components [15]), short circulation time (about 1 min at the size 3–5 µm [16,17]), and mechanical deformation capacity [18,19] due to the absence of a rigid core (important for maintaining blood flow [20]). The last two properties make it possible to achieve an effective accumulation of microcapsules due to the effect of the first passage through the vessels of the target organ without disturbing blood supply during arterial delivery. The effectiveness and safety of the arterial targeting of microcapsules to the vessels of the mesentery [20] and the hind paw [14] have been demonstrated. It was previously shown [21] that injection of microcapsules into the renal artery of healthy mice allows for the localization of 20% of the model substance (albumin conjugated with a fluorescent dye) in the target kidney. Localization becomes possible with a correct administration protocol, suitable container type, and the right dosage delivered to the organ [22].

Polyelectrolyte microcapsules are suitable for the encapsulation of high molecular weight cargo: protein-like biologically active substances, such as NGF [23], mRNA [23], and siRNA [24]. This feature should be taken into account when choosing a therapeutic agent. In the case of successful accumulation of such microcapsules in the target organ, they become sources of release into the bloodstream of the organ and into the interstitium of the biologically active substance during biodegradation. The problem of effective localization in the region of interest is currently being solved by modifying the carrier surface with various vector molecules (antibodies, aptamers, DARPin) [25,26,27] or by retention of the carriers with magnetic nanoparticles impregnated into the structure under magnetic field [28]. The success of such methods is debated. An alternative is the use of endovascular surgery methods, namely the administration of drug delivery systems into the artery supplying the target organ, which leads to the passage of the entire injected dose through this organ. As noted earlier, this strategy significantly increases the proportion of drug delivery systems retained in the target organ and reduces side effects.

In our work, we proposed and tested a new concept of murine glomerulonephritis-targeted therapy based on advances in drug carrier arterial delivery and encapsulation of etanercept into polyelectrolyte microcapsules for prolonged release. Etanercept is an immunosuppressant (hybrid dimeric protein molecule, 75 kDa). It is a competitive inhibitor of the binding of TNF-α to its receptors on the cell surface and thus inhibits the biological activity of TNF-α. We also studied in detail the localization, biodistribution, accumulation, and migration of polyelectrolyte microcapsules with drugs in organs during intra-arterial and intravenous administration.

## 2. Results and Discussion

### 2.1. Conjugation of Etanercept with Cy7

To enable ex vivo and in vivo visualization of etanercept in free and encapsulated form, it was conjugated with the fluorescent dye Cyanine 7 activated ester (Cy7) according to the protocol for proteins [14] with further purification from the unreacted dye. Then, the solution was centrifuged in a concentration column with a pore size of 9K MWCO to determine the efficiency of etanercept conjugation with Cy7. Cy7 has a low molecular weight (733.64 g/mol), which allows the imposed molecules to pass through the filter pores. After centrifugation, the fluid that passed through the filter had no characteristic coloration or fluorescence around 650 to 1350 nm. The retentant had a bright green coloration, indicating the possible effectiveness of conjugation. 

The etanercept amount in the solution was determined according to Bradford with calibration for the pure substance; the linked amount of Cy7 after dialysis was determined from the absorption maximum at a wavelength of 748 nm. According to spectrophotometric analysis, the percentage of Cy7 conjugated with protein was 92.7% (404 μg). Later, during the formation of polyelectrolyte microcapsules, quantitative calculations were carried out indirectly for the encapsulated etanercept-Cy7 by the fluorescence of Cy7.

To confirm the presence of etanercept in the conjugate, an enzyme immunoassay (ELISA) was performed by antibody to tumor necrosis factor alpha (anti-TNFa) since etanercept consists of human tumor necrosis factor receptor and Fc-linked human immunoglobulin G1. The analysis was performed in a 96-well plate, adsorption was measured at 450 nm, and each subject was measured in three replicates. Cy7 absorption spectrum is not observed at 450 nm and, therefore, horseradish peroxidase ELISA can be used (Figure A1). The medical drug Enbrel (active ingredient - etanercept, 10 mg) with the excipients mannitol 40 mg, sucrose 10 mg, and trometamol (TRIS) 1.2 mg. It was compared with etanercept-Cy7 conjugate in 0.9% NaCl. The test was performed after 24 and 72 h of storage. The conjugate was shown to contain the drug after 24 h of storage. After 72 h, the drug was not detected by ELISA, presumably due to degradation. 

The short lifetime of the etanercept molecule (no more than 72 h) is important to consider when planning experiments. Other studies show that the concentration of etanercept solution also has a significant effect on its stability over time [29]. In this regard, we tried to ensure that the time between the preparation of the etanercept solution and its direct injection into animals in its free and encapsulated form did not exceed 24 h.

### 2.2. Synthesis and Characterization of Microcapsules with Etanercept and Etanercept-Cy7

The encapsulation of the Cy7-conjugated etanercept was carried out by the previously developed method for the co-precipitation of high-molecular-weight proteins into the CaCO_3_ pores [30]. The main scheme of the formation of the polyelectrolyte microcapsules containing etanercept-Cy7 is shown in Figure 1. Polyelectrolyte microcapsules were obtained by LbL adsorption of oppositely charged biopolymers—polyarginine (pArg) and sodium dextran sulfate (DsS) on the surface of CaCO_3_ core containing etanercept-Cy7. After the adsorption of 3 pArg/DsS bilayers, the core was dissolved using an EDTA solution (0.2 M, pH 7.4). At each stage of the formation of polyelectrolyte layers and core, dissolving supernatant suspensions were taken to determine the amount of washed-out dye. In view of the fact that the Bradford reaction is based on the formation of a colored Coomassie Blue complex with free NH_2_-groups of amino acid residues, including arginine, it is not possible to determine the etanercept released from capsules during their formation [31]. In this regard, etanercept and a fluorescent dye in the near IR range (Cy7) were conjugated. Neither dextran sulfate nor poly-L-arginine has native fluorescence peaks in this range. Thus, the determination of the encapsulated etanercept-Cy7 by measuring the fluorescence intensity at a wavelength of 776 nm of the released dye in the supernatants, although indirect, makes it possible to isolate etanercept from other molecules in the solution. We consider the bond formed between etanercept and Cy7 during the conjugation process to be strong and thus neglect the spontaneous transitions of Cy7 to other molecules that make up the capsule shell. The final content of the dye in the microcapsules was 36.12 ± 0.09 μg, which corresponds to 58.1% of the initial amount of the dye.

Similar polyelectrolyte microcapsules containing only etanercept were synthesized to confirm the preservation of functionality and to perform a comparative ELISA analysis against anti-TNFa etanercept-Cy7 encapsulated in carriers. SEM microscopy shows that the capsules are morphologically similar to each other (Figure 1B,C) and did not contain a solid CaCO_3_ core. The similarity of the two types of capsules is confirmed by the size (Figure 1E,F). The mean diameters of microcapsules containing etanercept and etanercept-Cy7 were 2.65 ± 0.52 μm and 2.79 ± 0.44 μm, respectively. CLSM images of hollow microcapsules containing etanercept-Cy7 show that the main part of etanercept-Cy7 is linked with the microcapsule’s shell (Figure 1D). The adsorption of the negatively charged etanercept molecule [29] on the inner surface of the microcapsule after the core dissolution is explained by electrostatic interaction with the positively charged polyarginine [32] in the first layer of the shell. There are no significant differences in microcapsule ζ-potential: −26 ± 1 mV for encapsulated etanercept and −24 ± 1 mV for etanercept-Cy7, which suggests the colloidal stability of the microcapsules. 

ELISA performed the primary evaluation of etanercept and etanercept-Cy7 excretion from the microcapsules. For this purpose, microcapsules of the composition (polyarginine/dextran sulfate)_3_ with etanercept or etanercept-Cy7 incorporated were incubated for 24, 48, and 72 h at 36 °C in three different media: phosphate salt buffer (PBS) pH 7.2, NaCl 0.9% pH 7, and blood plasma pH 7.4. After incubation, the microcapsules were precipitated at 500 rpm for 2 min, and the supernatant was analyzed. The addition of etanercept-Cy7 microcapsules to the media resulted in a pH shift to an alkaline environment. According to enzyme immunoassay data, the largest amount of the drug came out in plasma samples containing 18 ng/mL of active ingredient etanercept or 108 ng/mL of Enbrel. Moreover, etanercept-Cy7 excreted better than etanercept, which may be due to a more cardinal shift in pH (Table A1) [33]. As is known, the buffer capacity of polyelectrolytes depends on the composition of the environment in which they are located [34,35]. Notably, the value of the buffer capacity and permeability of polyelectrolyte microcapsules is the sum of the entire shell, and not just its surface. The observed change in the pH (Table A1) indicates a low buffer capacity of polyelectrolyte microcapsules in general. Microcapsules formed by three pArg/DsS bilayers have a high penetrating ability, and an increase in pH leads to increasing the shell permeability [33,36] of the microcapsule shell and an increase in the released protein. The presence of etanercept and etanercept-Cy7 in the supernatant in PBS and NaCl was super low. 

Since the ability of labeled etanercept to bind specifically even after being encapsulated in polyelectrolyte layers was confirmed, only labeled etanercept capsules were used in further studies.

### 2.3. Changes of Kidney Blood Flow Associated with Microcapsules Intra-Arterial Administration

Microcapsules administration via intra-arterial injection was chosen as the targeting method to ensure local delivery of etanercept on both kidneys. To do this, the catheter was inserted through a puncture into the femoral artery, then into the abdominal aorta (Figure 2A). When the catheter tip was located between the branches of the right and left renal arteries, the abdominal aorta was occluded above and below the renal arteries using vascular clips. The total time of abdominal aorta occlusion was no more than 2 min. Prolonged occlusion (tens of minutes) of the aorta can lead to ischemic kidney damage [37]. Occlusion was performed to minimize the possibility of microcapsules movement down the abdominal aorta and their rapid systemic dispersal to other organs and tissues. In the proposed case, at the time of injection, most of the capsules enter the renal arteries due to blockage of blood flow in the aorta.

Monitoring of renal blood flow was performed to study the safety of intra-arterial injection of encapsulated immunosuppressant. The dosage of 22 × 10^6^ capsules containing 100 µg of etanercept-Cy7 in 50 µL of saline was chosen according to literature data [38,39]. Based on our previous experience with capsules of this type [22], that dosage of microcapsules with a size of 2.79 µm should have been safe for the vessels and tissues of the kidneys. In this regard, we only confirmed our assumption. The results of the study using laser speckle contrast imaging (Figure 2B) showed a decrease in blood flow 15 min after microcapsules’ injection (no more than 40% of the basal value) and complete recovery to the basal level after 24 h. Histological analysis also did not reveal any damage to the structure of renal tissue 5 days after the microcapsules’ injection (Figure A2).

In early studies, it was shown in detail that blood flow monitoring to assess the reversibility/irreversibility of embolization of the vital organs’ vessels after the drug delivery systems injection is extremely effective and important [16]. Blood flow measurement provides data on the state of the near-surface (peritubular) capillaries of the kidney (approximately 50–70 µm in depth), which originate from the efferent arterioles of the glomeruli. In previous work, we have also shown that a decrease in blood flow in the peritubular capillaries is an indicator of blockage of the glomerular capillaries located at a depth of 400–900 µm from the kidney surface [40]. This data on the restoration/non-restoration of blood flow to basal values fully correlated with data on the morphological state of the renal tissue. Thus, we showed that the injection of capsules based on polyarginine and dextran sulfate (2.88 μm in diameter) at a dose of 20 × 10^6^ capsules in 10 μL of saline through the left renal artery did not cause irreversible changes in blood flow and tissue morphology in the target kidney [21]. At the same time, the concentration had to be significantly reduced to 10 × 10^6^ capsules in 10 μL of saline for capsules of larger diameter (3.9–4 μm) [22,40]. 

### 2.4. Biodistribution Kinetics of the Etanercept-Cy7 after Microcapsules Intra-Arterial or Tail Vein Administration

The efficiency of accumulation of microcapsules with an immunosuppressant (etanercept labeled with Cy7) was studied using a model of an acute inflammatory process in the renal tissue induced by injection of 20 µL of a complete Freund’s adjuvant under the capsule of the left kidney 2.5 months before the start of the experiment. The inflammation of the kidney parenchyma picture is typical for this model (Figure A3). Active migration of leukocytes into the organ, as a response to the action of Freund’s adjuvant, creates large foci of leukocyte infiltrates under the kidneys’ capsule (Figure A3A). Extensive local infiltrates may form at the injection site if the adjuvant enters the kidney parenchyma, which is not a generalized process for the organ but remains a local injury (Figure A3B). The organ has a vascular plethora (Figure A3C), and the glomeruli have hypercellularity (Figure A3D). Furthermore, for this model, in addition to the initial stages of the formation of glomerulonephritis, signs of tubular nephritis are characteristic since many tubules have epithelial degeneration or occlusion of the lumen (Figure A3A,C,D). 

Fluorescent biodistribution analysis in vivo 1 min after administration of microcapsules intravenously (Figure 3A) or intra-arterially (Figure 3B) revealed significant differences between the two injection methods, which were fully consistent with ex vivo data.

According to ex vivo analysis, the intravenous administration was accompanied by the accumulation of most of the fluorescent signal in the lungs (about 60%) 1 min after injection (Figure 3C). However, after 1 h, there was a pronounced migration of the fluorescent signal to the liver (up to 72%) with a decrease in the signal from the lungs (up to 15%). After 24 h, there was a tendency to redistribute the fluorescent signal between the liver and intestines (up to 63% and 20% in the liver and intestine, respectively). Such a trend is typical for microcapsules 3–5 μm in size. This trend is associated with the primary sticking of microcapsules in the vessels of the lungs as a result of the first passage through them and subsequent gradual migration to the liver. After the capsules enter the liver, a slow release of the fluorescent cargo (etanercept-Cy7) begins under the action of the enzymatic system of hepatocytes and decomposition in Kupffer cells, followed by the subsequent excretion of metabolic products into the intestine with bile. However, at all time points, the level of the fluorescent signal in both kidneys did not exceed 1–5%. 

On the contrary, with the introduction of microcapsules into the renal arteries after 1 min, the accumulation of the fluorescent signal in the kidneys was 18%, against 5%, typical for intravenous administration. Additionally, a high level of fluorescent signal was observed in the intestine (50%), which is probably associated with the entry of capsules into the mesenteric vessels through the branches of the renal arteries. Signal accumulation in the kidneys and intestines was also visualized well in vivo. In the lungs and liver, the signal was much lower than with intravenous administration at the same time point (13 and 18%, respectively). By the first hour of the experiment, however, only 8% of the fluorescent signal remained in the kidneys according to tomography, which, nevertheless, was four times higher than with intravenous administration at the same time point. It was observed that 24 h after intra-arterial injection of microcapsules, the trend of fluorescent signal biodistribution in the organs of mice with pathology was generally similar to the trend noted after intravenous administration, while the signal level in the kidneys remained at least two times higher.

Data on the dynamics of the polyelectrolyte microcapsules’ biodistribution after intravenous administration are consistent with the results presented in other research papers obtained using MRI and classical histological analysis [15] and fluorescence tomography [14,17,41]. Arterial delivery contributed to a significant redistribution of microcapsules in organs, significantly increasing the efficiency of accumulation in the area of interest (Figure 3). This result can be achieved by using the effect of the first passage through the vessels supplying the area of interest. 

### 2.5. Capsules’ Localization and Following Elimination from Kidneys after Microcapsules Intra-Arterial or Tail Vein Administration

To assess the localization of microcapsules containing etanercept in kidney tissues, the thin tissue sections were examined using confocal laser scanning microscopy (CLSM). For this, the microcapsules were additionally labeled with a conjugate of bovine serum albumin with rhodamine B isothiocyanate (BSA-RITC). Microcapsule injection was carried out through the tail vein and renal arteries (see Figure 2A). The kidneys were removed 1 min and 1 h after the injection of the capsules and placed in a 10% formalin solution. Thin tissue sections were obtained with a cryostat. Thus, prior to the procedure, the organs were sequentially placed in phosphate-buffered saline (1 week) and sucrose solutions (10%, 20%, 30% for 2 h at each concentration). For analysis, a longitudinal section 70 μm thick from the central part of the right and left kidneys was used. Thus, all characteristic structures of the cortical layer and medulla of the kidney were presented on the sections, such as glomeruli, proximal and distal convoluted tubules, and collecting ducts. Since the microcapsules are several times smaller than the thickness of the tissue cryosection, to obtain the localization of all capsules present in the section, 3D scanning was carried out using CLSM. To visualize microcapsules, the maximum intensity projection of RITC-labeled objects’ three-dimensional distribution was calculated (Figure 4, green color). For the tissue autofluorescence channel, one of the XY planes from the 3D scan was chosen, where all structures of the kidney section were in the best focus. The resulting images are shown in Figure 4.

At intravenous administration of capsules, after 1 min, there are only a few capsules observed (1–2 per section), while after 1 h they are completely absent (Figure 4, bottom row). With intra-arterial administration, microcapsules were localized mainly in the glomeruli of the kidney after 1 min. For tissue samples taken 1 h after the administration of the capsules, a decrease in the number of capsules in the section is noticed; however, the remaining number was markedly higher than 1 min after intravenous administration.

It should be noted that the introduction of capsules into both renal arteries through the abdominal aorta (Figure 2A) was followed by the accumulation of a similar number of capsules in the left and right kidneys (according to fluorescence tomography, Figure 3B) and their similar localization (according to CLSM data, Figure 4) for the same time points.

The predominant accumulation of microcapsules in the kidney glomeruli is largely due to the complex structure of their capillary network [42]. The microcapsules most likely become mechanically stuck in glomerular capillaries due to the presence of a large number of bifurcations in conjunction with low blood flow (2–4 mm/s [43]). At the same time, the ability of the microcapsules to change shape due to the absence of a rigid core [18,44,45] most likely ensures their gradual rewashing from the capillaries. With regard to this process, there is a significant decrease in the number of microcapsules in the glomeruli after 1 h (Figure 4). Removal of microcapsules from the glomerular capillary network during the first hours is also a necessary condition for ensuring the safety of targeted drug delivery to the kidney [21].

### 2.6. Kinetics of the Fluorescent Marker Accumulation in Kidneys after Intra-Arterial Administration of Etanercept-Cy7 in Free and Encapsulated Form

To further study the biodistribution, the dynamics of changes in the accumulation of free and encapsulated etanercept in the kidneys when administered into the renal arteries were investigated using spectrophotometry. To conduct this experiment, the previously published [21] method of enzymatic destruction of mouse organs with trypsin (10 mg/mL in 50 mM Tris-HCl buffer solution) was carried out. For calibration, intact kidneys were used, to which known concentrations of free or encapsulated Cy7-labeled etanercept were added before the trypsinization procedure. Trypsinization was carried out separately for a series of organs containing free etanercept and an encapsulated form of the substance. The fluorescence spectra of obtained supernatant after trypsin solution action on the kidneys were recorded (excitation at 745 nm, detection at 770–850 nm). The amount of Cy7-labeled etanercept in organs was assessed at a wavelength of 785 nm. 

The administration of microcapsules into the renal arteries was accompanied by an accumulation of an average of 18, 8, 2.5, and 3% of the dose in the kidneys after 1 min, 1, 3, and 24 h, respectively (Figure 5). These data were similar to those obtained using a fluorescent tomograph (Figure 3C). Concurrently, the administration of labeled etanercept in free form led to the accumulation of only 2.5% of the dose after 1 min and 1 h, with an increase to 5% by 3 h (which is most likely due to drug metabolism) and a subsequent decrease by 24 h.

Overall, the data supported the efficacy of arterial targeting of the encapsulated form of etanercept compared to the free form and to intravenous administration of this type of container. However, it should be mentioned that the local effect of the drug on the renal tissue will most likely last no longer than 1–2 h. Therefore, it is better to adapt the cargo release profile from the capsules to this time interval.

### 2.7. Therapeutic Effect after Intra-Arterial Administration of Etanercept-Cy7 in Free and Encapsulated Form

Intra-arterial administration of etanercept solution did not cause significant changes in the morphological state of the renal tissue (Figure 6A) in comparison with animals not subjected to therapy (Figure A3). In this group of animals, all pathological changes in the renal tissue characteristic of this model of the disease were represented: hypercellularity of mouse kidney glomeruli, signs of tubular nephritis, degeneration of glomerular capillaries, and collapse of the Bowman’s capsule. Animals treated with an intra-arterial injection of etanercept microcapsules (Figure 6B) demonstrated similar pathological changes, but their severity was significantly lower. This may indicate the effectiveness of this route of administration of the encapsulated immunosuppressant. We probably did not observe a complete recovery of the kidney tissue due to significant morphological changes caused by the adjuvant injection at the time of the start of therapy. 

As the criteria for assessing the therapeutic effect of etanercept, morphological analysis of tissues was chosen as the most revealing. Other types of studies (assessment of the amount of protein in the urine and the intensity of diuresis) demonstrated more controversial differences between mice within groups. Most likely, this was due to the initial differences in the degree of disease progression in all animals at the time of the start of the experiment and individual response to the presence of an adjuvant in renal tissues. The therapeutic effect of the components of the capsule shell (dextran sulfate and polyarginine) was not expected. So, the group of animals that would have been injected with pristine capsules was not considered. Positive dynamics after the introduction of capsules with etanercept indicate that a safe dosage of capsules has been selected. 

Based on the obtained results, we assume that beginning the therapy at the earlier stages of the inflammatory process in the kidneys can have a much more pronounced protective effect on the developed approach.

## 3. Materials and Methods

### 3.1. Materials 

Calcium chloride, sodium carbonate, ethylenediaminetetraacetic acid disodium salt, dextran sulfate sodium salt (MW > 70,000), poly-L-arginine hydrochloride (MW > 70,000), Rhodamine B isothiocyanate, phosphate-buffered saline (PBS, 0.01 M), bovine serum albumin, trypsin from porcine pancreas (∼1500 U/mg), and tris(hydroxymethyl)aminomethane (≥9.8%) were purchased from Sigma-Aldrich (Taufkirchen, Germany). Hydrochloric acid was obtained from Reakhim (Moscow, Russia); dimethyl sulfoxide was purchased from Merck (Darmstadt, Germany); cyanine 7 NHS ester (Cy7) was obtained from Lumiprobe (Moscow, Russia). All chemicals were used without further purification. Deionized water produced with a water treatment system Milli-Q (Merck Millipore, Darmstadt, Germany) was used in all experimental stages. ELISA Kit For Anti-Tumor Necrosis Factor Alpha Antibody (Anti-TNFa) AEA133Hu from Cloud-Clone Corp. (Katy, TX, USA), 9K MWCO concentrators Thermo Fisher Scientific (Waltham, MA, USA).

### 3.2. Instruments 

ClarioSTAR Plus (BMG Labtech, Ortenberg, Germany), Zetasizer Ultra Red Label (Malvern Panalytical, Malvern, UK), IVIS SpectrumCT In Vivo Imaging System (PerkinElmer, Waltham, MA, USA), Environmental shaker incubator ES-20/60 (BioSan, Riga, Latvia), TS-100 Thermo-Shaker (BioSan, Riga, Latvia), Leica CM1950 cryostat (Leica Biosystems, Wetzlar, Germany), Leica TCS SP8X CLSM (Leica Biosystems, Wetzlar, Germany).

### 3.3. Conjugation of Etanercept with Cy7

Conjugation of etanercept by Cy7 was carried out according to the protocol for obtaining BSA-Cy7 with minor changes [14]. Namely, 1 mL of water for injection was added to the “Enbrel” ampoule. The aliquot (400 μL) of the prepared solution of etanercept (25 mg/mL) was taken and adjusted to 5 mL by saline (2 mg/mL). 5 mL of a Cy7 solution (1 mg/mL, in DMSO) was added to the mixture with vigorous stirring. The resulting mixture was incubated at room temperature by stirring for 4 h. Non-performed low molecular weight substances, including additives to the commercial "Enbrel", were removed by centrifugation in a concentration column with a pore size of 9K MWCO. The amount of the conjugated Cy7 dye with etanercept was calculated from spectroscopy measurements. The amount of the etanercept was calculated from spectroscopy measurements according to Bradford protocol. 

### 3.4. Synthesis and Characterization of Microcapsules with Etanercept and Etanercept-Cy7

To prepare microcapsules containing etanercept or etanercept-Cy7, a protein encapsulation protocol by co-precipitation into CaCO_3_ vaterite microspheres was used [46]. CaCO_3_ cores were prepared by mixing equal amounts of solutions of CaCl_2_ and Na_2_CO_3_ salts (0.615 mL of 1 M aqueous solutions) with the addition of 1.5 mL of ultrapure water and 1 mL of etanercept/etanercept-Cy7 solution (2 mg/mL). CaCO_3_ crystals containing etanercept/etanercept-Cy7 were precipitated by centrifugation at 1000 rpm and washed three times with ultrapure water. The prepared templates were used as templates for the layer-by-layer formation of microcapsules. To do this, sequentially applied layers of positively and negatively charged polyelectrolytes—pArg and DsS with triple washing of each layer with ultrapure water. Upon reaching three bilayers (the shell composition is pArg/DsS/pArg/DsS/pArg/DsS), the vaterite cores were dissolved in 0.2 M EDTA (pH 7.3) to form hollow microcapsules. The amount of the remaining Cy7 dye in the capsules was calculated from spectroscopy measurements by the analysis of the dye remaining in the supernatants after each core washing step during the encapsulation process.

### 3.5. Animal Studies

The experiments on animals were carried out at the Ogarev Mordovia State University. All in vivo and ex vivo procedures were treated according to the rules of the University Ethics Committee (Protocol No. 100, dated 29 November 2021, Saransk, Russia) and the Geneva Convention of 1985 (International Guiding Principles for Biomedical Research Involving Animals). The experiments were performed on Balb/c female mice 8 weeks old (25 g weight) using general anesthesia via intraperitoneal injection (Zoletil mixture (40 mg per kg, 50 μL, Virbac SA, Carros, France) and 2% Rometar (10 μL and 10 mg per kg, Spofa, Czech Republic)). The animals were euthanized by an overdose of anesthesia at the end of the experiments.

#### 3.5.1. Capsules’ Administration

The injection via intra-arterial injection was chosen as the targeting method to ensure local delivery of etanercept in free and encapsulated form on both kidneys. The catheter (5 mm, PU tubing, 32ga/0.8Fr, 0.005 × 0.010 in, Instech Laboratories, Inc., Plymouth Meeting, PA, USA) was inserted through a puncture into the femoral artery, then into the abdominal aorta (Figure 2A). When the catheter tip was located between the branches of the right and left renal arteries, the abdominal aorta was occluded above and below the renal arteries using vascular clips. Microcapsules (at a dose of 22 × 10^6^) were administered in 50 µL of saline. The clips were removed after 1 min, and the catheter was pulled out of the arteries.

#### 3.5.2. Blood Flow Response to Capsules’ Administration

A self-made laser speckle contrast imaging (LSCI) system [22] was used to monitor in vivo the effect of surgery and injection of a suspension of microcapsules through the renal arteries on the superficial blood supply to the kidneys. LSCI is a non-destructive optical method for measuring superficial blood flow in vessels. The LSCI system is equipped with a 635 nm diode source (CPS635S, Thorlabs, Newton, NJ, USA) and digital camera (a2A2600-64ucBAS, Basler AG, Ahrensburg, Germany). LSCI is a non-contact and non-destructive optical modality that allows for qualitative measurement of blood flow in superficial vessels [47,48,49,50].

Mice were divided into two groups: with (experimental) and without (control) microcapsules administration. To access the surface of the kidney, a small incision was made in the region of the left kidney parallel to the spine of mice. Before measurements, the surrounding tissues were moved aside and fixed. Cutting the incision, renal artery catheterization, and measurement were performed under general anesthesia. 

Speckle-contrast data were recorded directly from the surface of the left kidney before, 15 min, 1 h, and 24 h after microcapsules’ injection into the renal artery. For each mouse, speckle contrasts measured from each kidney before the injection were considered basal. Contrast data measurements made after were normalized to the basal values for the specific kidney of each specific mouse. 

The exposure time of the digital camera was set to 10 ms, and the frame rate was set to 30 frames per second. To increase the signal-to-noise ratio of the computed speckle contrast images, 25 consecutive speckle contrast frames were averaged. For each dataset, an ROI of 250 × 250 pixels was selected; the mean and standard deviation were calculated in the ROI.

#### 3.5.3. Microcapsules’ Biodistribution

The IVIS SpectrumCT In Vivo Imaging System (PerkinElmer, Waltham, MA, USA) was used to measure the fluorescent marker Cy7 distribution in the mouse body after tail vein and intra-arterial injection (excitation and emission were set at 745 and 800 nm, respectively). Post-processing was performed via Living Image software v.4.7.3. The kinetics of the fluorescence redistribution of fluorescence signals in mouse organs (kidneys, lungs, heart, liver, spleen, stomach, intestines, appendix) were observed at 15 min, 1, and 24 h after 22 × 10^6^ microcapsules injection. 

#### 3.5.4. Microcapsules’ Localization in the Kidney Tissue

Prior to the localization study, microcapsules with encapsulated etanercept were additionally labeled with a conjugate of BSA with rhodamine B isothiocyanate (BSA-RITC). Both kidneys were collected after 1 min and 1 h after microcapsules injection through either tail vein or renal arteries (Figure 2A). Organs were placed in a 10% formalin solution. Then, kidneys were put sequentially in phosphate-buffered saline (1 week), and sucrose solutions (10, 20, 30 *w*/*v*% in water for 2 h at each concentration). Tissue thin sections (70 μm) were obtained from the central parts of both left and right kidneys with a Leica cm1950 cryostat (Leica Biosystems, Wetzlar, Germany).

Tissue samples were placed between two cover glasses for further CLSM measurements. Leica TCS SP8X (Leica Biosystems, Wetzlar, Germany) equipped with a 405 nm diode and 514 nm argon laser sources was used. Sample fluorescence was excited through the 20×/0.70 N.A. objective. The following detection ranges were used: excitation at 405 nm, detection at 420–500 nm (renal tissue autofluorescence); excitation at 514 nm, detection at 550–650 nm (BSA-FITC-labeled microcapsules). For each tissue sample, the complex CLSM measurements were performed: 4 × 1 viewfields (single viewfield size, 775 × 775 μm) with a little overlapping were registered with 3D scanning carried out in each viewfield. Leica software was used to perform mosaic stitching procedures on obtained data. The size of the area studied was 775 × 3100 × 50 μm for each sample. To visualize all microcapsules present in the measured area, maximum intensity projection of the 3D channel corresponding to BSA-FITC-labeled microcapsules was carried out.

#### 3.5.5. Quantification of Fluorescent Dye Content in Kidneys

To estimate the amount of either free or encapsulated Cy7-labeled etanercept contained in the organ at different times after injection through renal arteries, enzymatic destruction of kidneys was performed according to a previously published protocol [21]. Briefly, 1 mL of trypsin solution in Tris-HCl buffer (10 mg/mL, 50 mM buffer, pH 8.0) was added to pre-grinded organs. The mixture was left under constant temperature (37 °C) and shaking (1000 rpm) for 2 h and then centrifuged (6000 rpm, 1 min). 100 µL of supernatant (5 replicates) of each sample was placed in a 96-well plate.

The intact organs were used for calibration purposes. The known amount of either free or encapsulated etanercept was added to intact kidneys prior to the enzymatic destruction procedure. 

Fluorescence spectra of obtained supernatants were recorded with ClarioSTAR Plus (BMG Labtech, Ortenberg, Germany) with excitation at 745 nm and detection at 770–850 nm (step size, 1 nm). The amount of Cy7-labeled etanercept was evaluated for detection at 785 nm. 

#### 3.5.6. Modeling of Acute Glomerulonephritis

The model of acute glomerulonephritis in the renal tissue was induced by injection of 20 µL of complete Freund’s adjuvant under the capsule of the left kidney. 2.5 months after injection, when the inflammation of the kidney parenchyma occurs (Figure A3A), the therapy was started using etanercept in free and encapsulated form. 

#### 3.5.7. Histological Studies

Kidneys were fixed in neutral formalin and were then desiccated by dehydration in isopropyl alcohol and embedded in paraffin. The 5 μm-thick slides were stained with hematoxylin and eosin (H and E staining). Morphologic analysis of the histological samples was performed using an Olympus digital image analysis system.

## 4. Conclusions

The need to localize immunosuppressants as much as possible in renal therapy during glomerulonephritis therapy has been conditioned with a wide range of side effects after systemic administration to the body. In the present study, we have tested the concept of acute glomerulonephritis targeted therapy using the arterial delivery of etanercept in the capsulated and free form in a mouse model. In the first stage, we found that the capsules of 2.79 ± 0.44 μm at a dose of 22 × 10^6^ in 50 μL of saline (containing 100 μg of etanercept) decreased renal blood flow 15 min after intra-arterial administration. At the same time, by 24 h, there was a complete restoration of blood flow parameters to the basal level. Histological analysis also did not reveal any damage to the renal tissue structure. Arterial targeting of encapsulated etanercept has been shown to be effective in comparison to intravenous administration as well as arterial administration of etanercept in free form. The administration of microcapsules into the renal arteries was accompanied by an accumulation of an average of about 18 and 8% of the dose in the kidneys after 1 min and 1 h, respectively, while in the next 24 h, the number of capsules did not exceed 3%. Removal of the majority of microcapsules from the kidney within 24 h is an extremely important mechanism for maintaining their functioning. A detailed analysis of kidney tissues showed the presence of single capsules in sections 1 min after their injection into the tail vein and complete absence after 1 h. When injected into both renal arteries through the abdominal aorta, clearly distinguishable accumulations of capsules were noted in the left and right kidneys for 1 min. At the same time, microcapsules were mainly localized in the glomeruli of the kidney. One hour after administration, the number of capsules in the glomeruli decreased significantly. Histological analysis data confirmed the positive dynamics in acute glomerulonephritis therapy only in animals with intra-arterial administration of etanercept in encapsulated form. 

Generally, the developed therapeutic strategy is likely to be effective for different kidney diseases induced by the development of the inflammatory process in glomeruli. The described approach could be effective for the delivery of a wide range of high-molecular cargoes such as proteins, peptides, ferments, and genetic material.

## Figures and Tables

**Figure 1 ijms-24-02784-f001:**
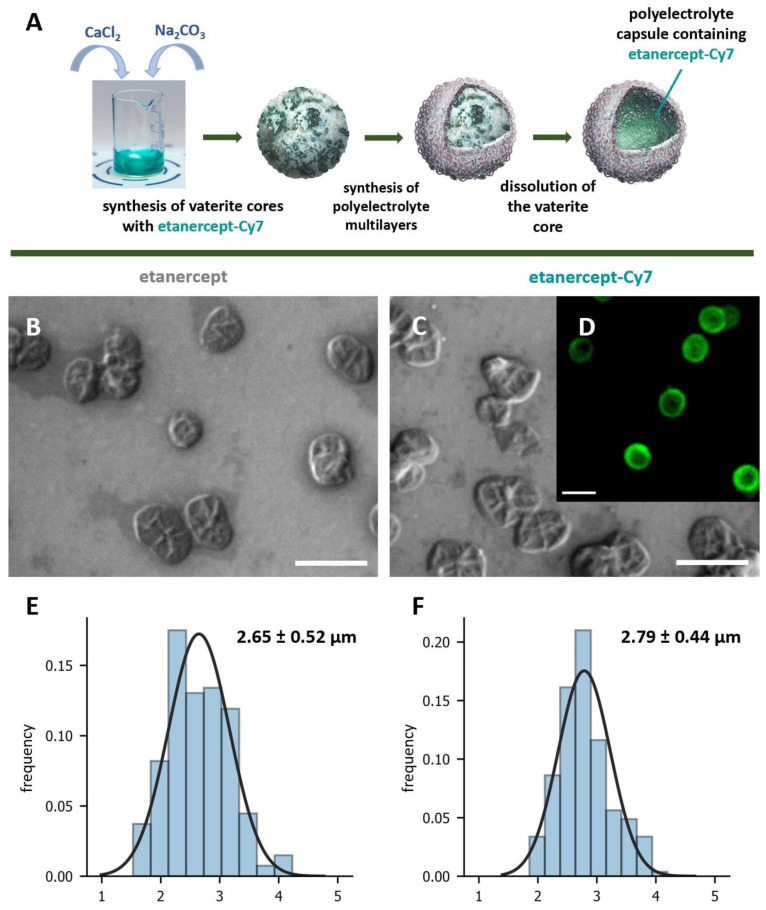
Scheme of etanercept-Cy7 encapsulation in polyelectrolyte microcapsules (**A**). Typical SEM images of microcapsules with etanercept (**B**) and etanercept-Cy7 (**C**). CLSM image of typical polyelectrolyte microcapsules with etanercept-Cy7 (**D**). Size distribution with Gaussian fit of microcapsules with etanercept (**E**) and etanercept-Cy7 (**F**) in aqueous solution. The diameter was determined from SEM images of 200 microcapsules. The scale bar is 5 µm.

**Figure 2 ijms-24-02784-f002:**
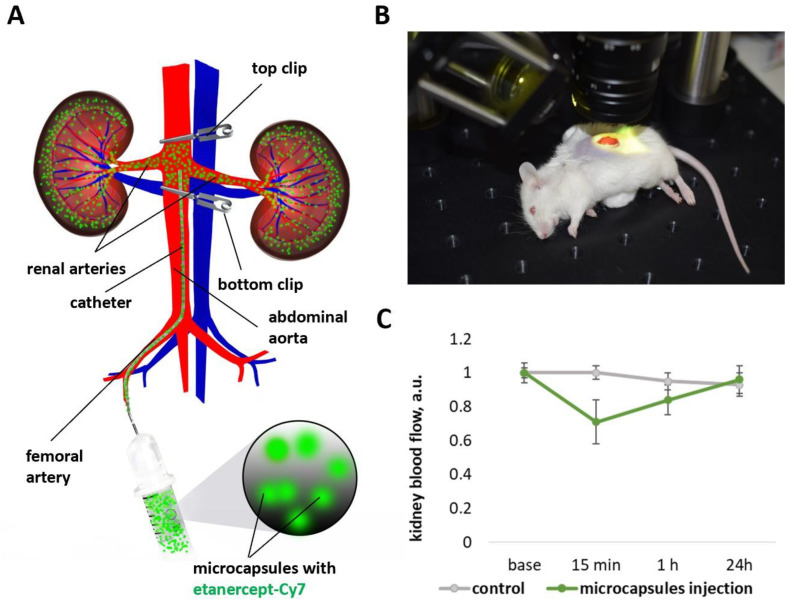
General scheme of the microcapsules injection into the renal arteries through the abdominal aorta (**A**). Measurement of mouse kidney blood flow using a laser speckle imaging system (**B**). Dynamics of kidney blood flow before, 15 min, 1 h, and 24 h after the microcapsules’ intra-arterial injection (**C**).

**Figure 3 ijms-24-02784-f003:**
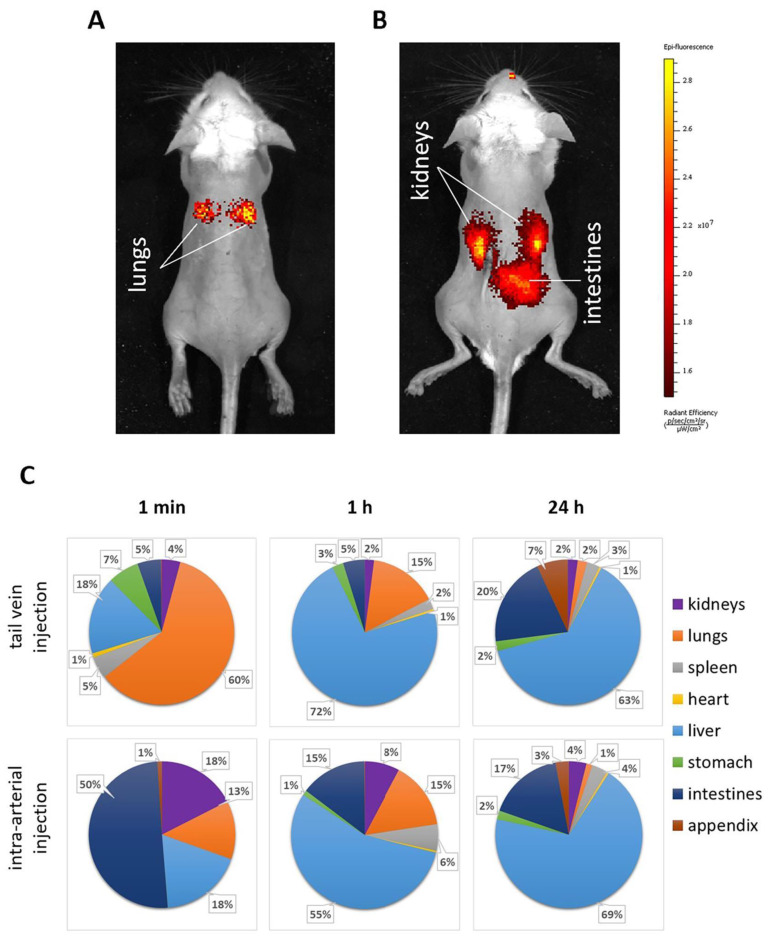
Biodistribution of the fluorescent signal in vivo (**A**) at 1 min, tail vein injection; (**B**) at 1 min, renal arteries injection) and ex vivo (**C**) in dynamics) after injection of microcapsules with etanercept-Cy7 via the tail vein or renal arteries through the abdominal aorta (intra-arterial injection). Negligible (<1%) ex vivo fluorescent signal (**C**) was not labeled.

**Figure 4 ijms-24-02784-f004:**
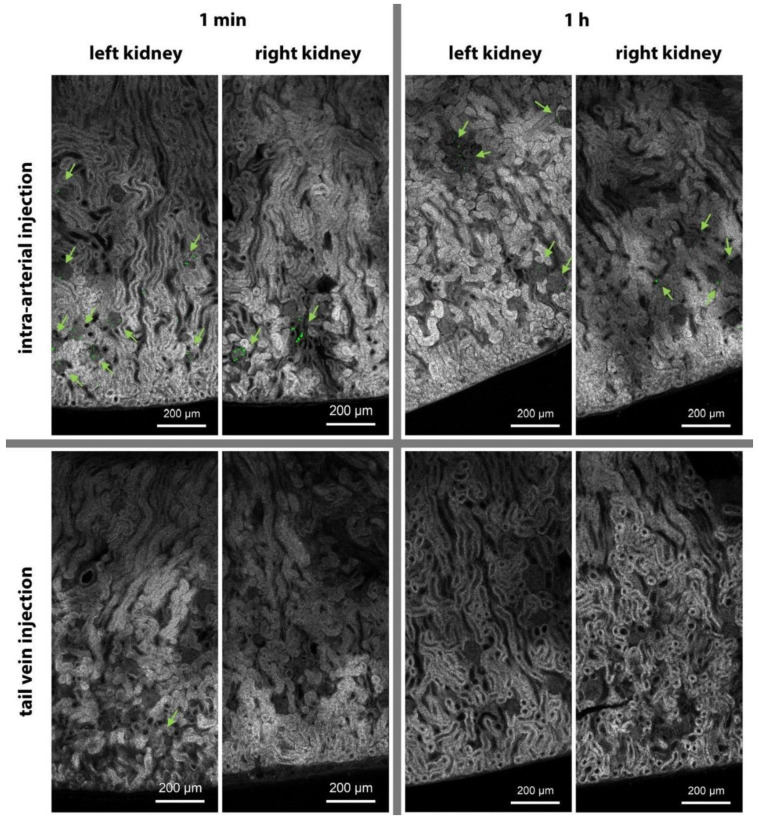
Results of 3D CLSM of kidney tissue cryosections (thickness 70 μm) after 1 min and 1 h with different methods of administration. Maximum intensity projection of fluorescent signal of microcapsules containing etanercept and labeled with BSA-RITC (green, 514 nm excitation, 550–650 nm detection). One of the XY planes with kidney tissue autofluorescence (gray, 405 nm excitation, 420–500 nm detection), in which all structures of the kidney section are in the best focus.

**Figure 5 ijms-24-02784-f005:**
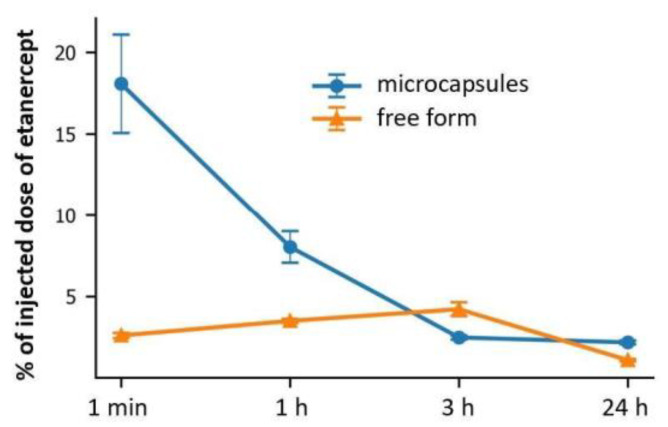
Dynamics of changes in the content of Cy7-labeled etanercept in encapsulated and free forms in the kidneys with their renal artery injection. The evaluation was carried out by fluorescence measurements of organ trypsinization products (excitation at 745 nm, detection at 785 nm).

**Figure 6 ijms-24-02784-f006:**
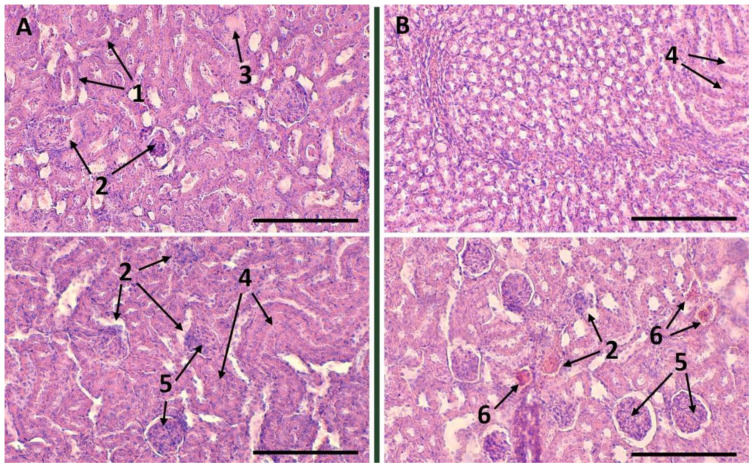
Renal tissue 7 d after therapy of acute glomerulonephritis using intra-arterial injection etanercept in free (**A**) and encapsulated (**B**) form: fibrin clots in the tubules’ lumen (1); degeneration of glomerular capillaries, collapse of the Bowman’s capsule (2); small foci of the renal parenchyma sclerosis (3); local accumulation of immune cells in the parenchyma (infiltrate) (3); epithelial dystrophy or tubular lumen occlusion (4); multicellularity and glomerular edema—a sign of inflammation, narrowing of the Bowman’s capsule lumen (5); vascular plethora (6). The thickness of the histological samples is 5 μm; dyes: hematoxylin and eosin. The scale bar is 200 µm.

## Data Availability

The data supporting the findings of this study are available from the corresponding author upon reasonable request.

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
