# Peer review of "Targeted Therapy for Glomerulonephritis Using Arterial Delivery of Encapsulated Etanercept"

_ijms, 2023, doi:10.3390/ijms24032784_

Round 1

Reviewer 1 Report

The article “Targeted therapy for glomerulonephritis using arterial delivery of encapsulated etanercept” by Natalia A. Shushunova et. al. is interesting and valuable work that makes a significant contribution to the development of immunosuppressive therapy. But at the same time, the article has several issues which are listed below:

  1. The article has a poor discussion part. It is important to discuss the effects demonstrated in that article to improve the scientific significance of the work.

  2. Also, it is unclear the novelty of the article's results. I will strongly recommend clarifying that point.

  3. The article is written in unclear and inaccessible language. The check of article by a native speaker is recommended.  

  4. The decision of the ethical commission about all experimental procedures is absent in the article. The only names of protocols are mentioned in the materials and methods part.

  5. In the abstract part, the size distribution of microcapsules is missed. 

  6. It is needed to clarify and simplify the abstract.

  7. “In this regard, the determination of the encapsulated etanercept-Cy7 was determined indirectly, by the evaluation of the amount of the released dye in the supernatants. “ Did it take into account the influence of polyarginine distribution during the dissociation of microcapsules? And how? 

  8. "CLSM  images of hollow microcapsules containing etanercept-Cy7 show that the protein cross-linked with the fluorescent dye is present in the entire volume of the carriers (Figure 1D)." Figure 1D demonstrated the opposite effect, the main part of etanercept-Cy7 linked with the microcapsule's shell. The discussion part is missed. 

  9. There are no control experiments with capsules without encapsulated etanercept.

  10. Do the authors intend to carry out the procedure for the introduction of capsules with the drug using arterial occlusion? Or would the introduction of polyelectrolyte microcapsules into the bloodstream without arterial occlusion affect the results?

  11. “etanercept-Cy7 excreted better than etanercept, which may be due to a more cardinal shift in pH and the dependence of dextran permeability from it (Table A1)” - It is necessary to prove, this since polyelectrolytes and their complex have their buffer capacity according to literature sources.

Author Response

1. The article has a poor discussion part. It is important to discuss the effects demonstrated in that article to improve the scientific significance of the work.

We have significantly expanded the discussion part to make our manuscript more useful and meaningful.

2. Also, it is unclear the novelty of the article's results. I will strongly recommend clarifying that point.

We thank the reviewer for this comment. The following text has been added to clarify the novelty of the research.

In the case of successful accumulation of such microcapsules in the target organ, they become sources of release into the bloodstream of the organ and into the interstitium of the biologically active substance during biodegradation. The problem of effective localization in the region of interest is currently being solved by modifying the carrier surface with various vector molecules (antibodies, aptamers, DARPin) [10.1016/S1359-6446(05)03698-6, 10.2174/0929867325666181008142831, 10.1134/S1068162020060308] or by retention of the carriers with magnetic nanoparticles impregnated into the structure under magnetic field [10.1016/j.jmmm.2010.11.058]. The success of such methods is debated. An alternative is the use of endovascular surgery methods, namely the administration of drug delivery systems into the artery supplying the target organ, which leads to the passage  through this organ of absolutely the entire injected dose. As noted earlier, this strategy significantly increases the proportion of drug delivery systems retained in the target organ and reduces side effects.

In our work, we proposed and tested the new concept of murine glomerulonephritis targeted therapy based on advances of the drug carrier arterial delivery and encapsulation of etanercept into polyelectrolyte microcapsules for prolonged release.” (pages 2, 3) 

3. The article is written in unclear and inaccessible language. The check of article by a native speaker is recommended.  

We have carefully checked the text of the manuscript and made corrections (all changes are marked in yellow).

4. The decision of the ethical commission about all experimental procedures is absent in the article. The only names of protocols are mentioned in the materials and methods part.

We have expanded the part with a decision of the ethical commission about all experimental procedures on animals in the "Materials and Methods" section. We have also added a "Institutional Review Board Statement" section in the manuscript (page 13). A copy of the decision of the Ethics Committee can be provided upon request (in local (Russian) language).

5. In the abstract part, the size distribution of microcapsules is missed. 

The size distribution of microcapsules was added to abstract.

6. It is needed to clarify and simplify the abstract.

The abstract was rephrased.

7. “In this regard, the determination of the encapsulated etanercept-Cy7 was determined indirectly, by the evaluation of the amount of the released dye in the supernatants. “ Did it take into account the influence of polyarginine distribution during the dissociation of microcapsules? And how? 

We rephrase the discussion to make make it more clear:

“In this regard, etanercept and a fluorescent dye in the near IR range (Cy7) were conjugated. Neither dextran sulfate nor poly-L-arginine have native fluorescence peaks in this range. Thus, the determination of the encapsulated etanercept-Cy7 by measuring the fluorescence intensity at a wavelength of 776 nm of the released dye in the supernatants, although indirect, makes it possible to isolate etanercept from other molecules in solution. We consider the bond formed between etanercept and Cy7 during the conjugation process as strong and neglect the spontaneous transitions of Cy7 to other molecules that make up the capsule shell.” (page 4)

8. "CLSM  images of hollow microcapsules containing etanercept-Cy7 show that the protein cross-linked with the fluorescent dye is present in the entire volume of the carriers (Figure 1D)." Figure 1D demonstrated the opposite effect, the main part of etanercept-Cy7 linked with the microcapsule's shell. The discussion part is missed. 

We have corrected the text as follows: “CLSM images of hollow microcapsules containing etanercept-Cy7 show that the main part of etanercept-Cy7 linked with the microcapsule's shell (Figure 1D). The adsorption of the negatively charged etanercept molecule [10.1016/j.ijpharm.2013.11.019] on the inner surface of the microcapsule after the core dissolution is explained by electrostatic interaction with the positively charged polyarginine [10.1039/C5NR07665J] in the first layer of shell. ” (page 4)

9. There are no control experiments with capsules without encapsulated etanercept.

The therapeutic effect of the components of the capsule shell (dextran sulfate and poly-L-arginine) was not expected. So, the group of animals that would have been injected with pristine capsules was not considered. The issues of safety and efficacy of using polyelectrolyte microcapsules were discussed earlier and presented in [10.1016/j.jconrel.2020.11.051, 10.1364/BOE.430393, 10.3390/pharmaceutics14051056]. Positive dynamics after the introduction of capsules with etanercept indicate that a safe dosage of capsules has been selected. The corresponding comment has been added to the text of the article (page 11)

10. Do the authors intend to carry out the procedure for the introduction of capsules with the drug using arterial occlusion? Or would the introduction of polyelectrolyte microcapsules into the bloodstream without arterial occlusion affect the results?

Total time of abdominal aorta occlusion was no more than 2 minutes. Prolonged occlusion (tens of minutes) of the aorta can lead to ischemic kidney damage [10.1097/01.ASN.0000057858.45649.F7]. Occlusion was performed to minimize the possibility of microcapsules movement of  down the abdominal aorta and their rapid systemic dispersal to other organs and tissues. In the proposed case, at the time of injection, most of the capsules enter the renal arteries due to blockage of blood flow in the aorta. We have added an appropriate clarification to the text of the manuscript (pages 5, 6)

11. “etanercept-Cy7 excreted better than etanercept, which may be due to a more cardinal shift in pH and the dependence of dextran permeability from it (Table A1)” - It is necessary to prove, this since polyelectrolytes and their complex have their buffer capacity according to literature sources.

“Moreover, etanercept-Cy7 excreted better than etanercept, which may be due to a more cardinal shift in pH (Table A1) [10.1002/admi.201600273]. As is known, the buffer capacity of polyelectrolytes depends on the composition of the environment in which they are located [https://doi.org/10.3390/ijms23179917,  https://doi.org/10.3390/polym12030520]. Note, the value of the buffer capacity and permeability of polyelectrolyte microcapsules is the sum of the entire shell, and not just its surface. The observed change in the pH  (Table A1) indicates a low buffer capacity of polyelectrolyte microcapsules in general. Microcapsules formed by three pArg/DsS bilayers have a high penetrating ability, an increase in pH leads to increasing the shell permeability [10.1002/admi.201600273, 10.1021/la1018949] of the microcapsule shell and an increase in the released protein”. We have added an appropriate discussion to the manuscript (page 4).

Reviewer 2 Report

This paper compares cy7 labeled etanercept used to attempt to counter autoimmune damage to the kidney, with an encapsulated form.

The evidence for the efficacy of the encapsulated etanercept is thin, being entirely circumstancial. However, no great claims are made for it.

The manuscript desperately needs to be spell and grammar checked, with simple errors like "cristall" in place of crystal being common. There are too many to enumerate. It needs to be checked and rewritten.

There are 35 refs in the list, missing perhaps some of the literature, but there is a ref 46 in 3.4, and higher numbers elsewhere. Clearly, the refs are messed up.

Author Response

1. The evidence for the efficacy of the encapsulated etanercept is thin, being entirely circumstancial. However, no great claims are made for it.

As the criteria for assessing the therapeutic effect of etanercept, morphological analysis of tissues was chosen as the most revealing. Other types of studies (assessment of the amount of protein in the urine and the intensity of diuresis) demonstrated more controversial differences between mice within groups. Most likely, this was due to the initial differences in the degree of disease progression in all animals at the time of the start of the experiment and individual response to the presence of an adjuvant in renal tissues. Based on the obtained results, we assume that the beginning of therapy at earlier stages of the inflammatory process in the kidneys can have a much more pronounced protective effect of the developed approach. The corresponding discussion has been added to the text of the article (page 11,12).

2. The manuscript desperately needs to be spell and grammar checked, with simple errors like "cristall" in place of crystal being common. There are too many to enumerate. It needs to be checked and rewritten.

We have carefully checked and rewrote the text of the manuscript.

3. There are 35 refs in the list, missing perhaps some of the literature, but there is a ref 46 in 3.4, and higher numbers elsewhere. Clearly, the refs are messed up.

We have carefully checked the text and fixed errors in refs.